# Drug Delivery through Epidermal Tissue Cells by Functionalized Biosilica from Diatom Microalgae

**DOI:** 10.3390/md21080438

**Published:** 2023-08-03

**Authors:** Danilo Vona, Annarita Flemma, Francesca Piccapane, Pietro Cotugno, Stefania Roberta Cicco, Vincenza Armenise, Cesar Vicente-Garcia, Maria Michela Giangregorio, Giuseppe Procino, Roberta Ragni

**Affiliations:** 1Chemistry Department, University of Bari “Aldo Moro”, Via Orabona 4, I-70126 Bari, Italy; danilo.vona@uniba.it (D.V.); annarita.flemma@uniba.it (A.F.); pietro.cotugno@uniba.it (P.C.); vincenza.armenise@uniba.it (V.A.); cesar.vicentegarcia@uniba.it (C.V.-G.); 2Bioscience, Biotechnology and Biopharmaceutics Department, University of Bari “Aldo Moro”, Via Orabona 4, I-70126 Bari, Italy; francesca.piccapane@uniba.it; 3Institute for the Chemistry of Organometallic Compounds (ICCOM), Consiglio Nazionale delle Ricerche (CNR), Chemistry Department, Via Orabona 4, I-70126 Bari, Italy; cicco@ba.iccom.cnr.it; 4Institute of Nanotechnology (Nanotec), Consiglio Nazionale delle Ricerche (CNR), Chemistry Department, Via Orabona 4, I-70126 Bari, Italy; michelaria.giangregorio@nanotec.cnr.it

**Keywords:** diatom biosilica, organosilane, topical drug delivery, diatomaceous earth, trans-epidermal drug permeation

## Abstract

Diatom microalgae are a natural source of fossil biosilica shells, namely the diatomaceous earth (DE), abundantly available at low cost. High surface area, mesoporosity and biocompatibility, as well as the availability of a variety of approaches for surface chemical modification, make DE highly profitable as a nanostructured material for drug delivery applications. Despite this, the studies reported so far in the literature are generally limited to the development of biohybrid systems for drug delivery by oral or parenteral administration. Here we demonstrate the suitability of diatomaceous earth properly functionalized on the surface with *n*-octyl chains as an efficient system for local drug delivery to skin tissues. Naproxen was selected as a non-steroidal anti-inflammatory model drug for experiments performed both in vitro by immersion of the drug-loaded DE in an artificial sweat solution and, for the first time, by trans-epidermal drug permeation through a 3D-organotypic tissue that better mimics the in vivo permeation mechanism of drugs in human skin tissues. Octyl chains were demonstrated to both favour the DE adhesion onto porcine skin tissues and to control the gradual release and the trans-epidermal permeation of Naproxen within 24 h of the beginning of experiments. The evidence of the viability of human epithelial cells after permeation of the drug released from diatomaceous earth, also confirmed the biocompatibility with human skin of both Naproxen and mesoporous biosilica from diatom microalgae, disclosing promising applications of these drug-delivery systems for therapies of skin diseases.

## 1. Introduction

Nanostructured silica-based materials deserve great interest for a wide variety of applications ranging from catalysis to sensing, photonics and environmental science [1,2,3,4,5,6]. The structural mesoporosity and high surface area, combined with biocompatibility and availability of versatile chemical modification approaches, make these materials suitable for the design and production of a plethora of multifunctional systems, with special focus on application in biomedicine and drug delivery [7,8,9].

Among natural sources of biosilica, diatoms are unicellular photosynthetic micro-algae accounting for up to 40% of marine phytoplankton, and they are responsible for 20% of the oxygen production in marine ecosystems [1,10]. Diatoms absorb, accumulate and deposit ortho-silicic acid, generating highly porous silica cell walls, called frustules, whose structure has been optimized by nature to enclose the eukaryotic protoplasm and protect it from noxious ultraviolet wavelengths, exogenous viruses and predators [11,12]. Frustules have a hierarchical organization of silica layers, with intricated micro and nano-porous patterns, thus providing high mechanical resistance and surface area [1]. Moreover, diatom biosilica can be easily functionalized with a wide variety of tailored molecules depending on its specific application [13,14,15,16,17] and it represents an eco-friendly alternative to synthetic silica nanoparticles, whose preparation requires expensive and organic solvent-consuming experimental protocols [18]. Besides integer biosilica shells of living diatoms, a huge amount of fossil sediments, i.e., the diatomaceous earth (DE) or diatomite [8], is available at extremely low costs. DE purification is generally necessary before its use and, for this aim, acid-oxidative protocols are available not only to oxidize and remove the organic residual fossil matter, but also to activate the hydroxyl groups on the silica surface and to promote their reactivity for chemical functionalization [19,20].

In the last decades, diatomite has been investigated as a biocompatible mesoporous material for the delivery of non-steroidal anti-inflammatory drugs (NSAIDs), siRNA and antibiotics for oral or parenteral administration [8,19,21,22,23,24].

Physisorption or chemisorption are the main mechanisms of drug adhesion onto porous DE, whereas drug delivery occurs by subsequent concentration gradient-driven diffusion in physiological media [8,19]. The possibility of performing proper chemical modification of the diatomite surface is also advantageous to control the delivery rate, avoiding a too fast drug release [20,24], as well as to favour selective interactions of the delivery system with specific target tissues, avoiding side effects in non-target organs [2,20,23]. 

Although DE is regarded by the Food and Drug Administration (FDA) as a safe bioproduct for human oral consumption, the authorisation for its biomedical application by oral administration has not been released yet [8,19], mainly because of its slow biodegradability in biological media and accumulation in organs, as reported in experiments on rats’ liver, lung, and kidney [19,20].

On this ground, drug delivery by topical administration can be a valid alternative for skin or musculoskeletal diseases, avoiding the drawbacks observed for oral drug administration, such as gastric pH variation, drug concentration gradients in plasma and systemic toxicity [25,26,27].

Drug delivery across the skin is a challenging focus since the physiological role of this organ is to act as a barrier against the external environment. Skin protects organisms from viruses, bacteria and particles such as allergens, as it is composed of different layers, although the one working as a barrier is the outermost layer, the stratum corneum, made of 15–20 layers of dead and anucleate keratinocytes densely organized in a lipid extracellular matrix [28,29].

The stratum corneum is also a barrier for water, preventing dehydration, and it represents a bottleneck in the diffusion across the skin of substances with a molecular weight higher than 500 Da, such as a variety of drugs [25,28]. For most drugs, passive diffusion occurs via intercellular routes, but intracellular and appendageal routes, as well as diffusion mediated by protein transporters, are also possible [28]. However, the main drawback is to control the drug amount reaching systemic circulation [26].

Skin drug delivery systems based on mesoporous silica, metallic, solid lipid or polymeric nanoparticles and liposome carriers, have been tested so far [26,30,31,32,33,34] and although studies report that skin penetration of nanoparticles with a diameter higher than 20 nm is negligible, when the epidermal barrier is compromised, aged or diseased, there may be potential for nanoparticle penetration with possible concerns of very low biodegradability of nanocarriers and their retention in the reticuloendothelial system [28]. On this ground, exploitation of DE microparticles may prevent this issue since their size distribution is not compatible with the trans-epidermal pathways available for the released drug molecules.

Among model molecular active principles suitable to investigate drug delivery through skin, Naproxen (Nap) is a well-known non-steroidal anti-inflammatory (NSAID) drug. It is widely exploited to relieve symptoms of arthritis such as inflammation, swelling, stiffness, joint pain or muscle aches. However, chronic oral administration of this drug leads to peptic ulceration and gastritis as the main issues [34,35,36]. In general, efficient penetration through the skin is favoured for drugs having both hydrophilic and lipophilic structural features [26,34,37], since their penetration can occur both across the lipophilic matrix of stratum corneum via the intercellular route and across hydrophilic keratinocytes via the intracellular route. Penetration is mainly dependent on the drug hydrophilic–lipophilic balance [38]. Regarding Naproxen, the literature shows that, upon topical administration, its skin penetration is limited, allowing only 1–2% of drug bioavailability in serum as an effect of the presence of the hydrophilic carboxyl group and of the lipophilic main structural backbone of the drug [39].

In this framework, we investigate the suitability of a series of functionalized DE samples as scaffold materials for drug delivery, using Nap as the model drug. Samples were functionalized on the surface with different concentrations of *n*-octyl groups with the aim of investigating the effect of aliphatic chains on DE adhesion to the hydrophobic skin external layer.

Moreover, we demonstrate, for the first time, evidence that Naproxen delivered from DE microparticles can cross a human artificial skin insert (EpiDerm^TM^) without altering skin cell morphology and viability.

## 2. Results

With the aim of comparing and exploring the different hydrophobic interactions and adhesive properties on the lipidic matrix of skin tissues, a series of diatomite samples were functionalized on the surface with *n*-octyl-triethoxysilyl groups at different concentrations. First, they were prepared by a purification step of diatomaceous earth (DE) with an acid-oxidative protocol leading to activated diatomite (DA) and a subsequent condensation reaction, in toluene at 80 °C, of the hydroxyl groups of DA with triethoxy-octylsilane (TeC_8_S) used at 1, 5, 10, 20 and 40% (TeC_8_S:DA *w*/*w*).

The resulting functionalized diatomite samples (DF1, DF5, DF10, DF20 and DF40, respectively) were fully characterized by Fourier Transform InfraRed (FT-IR) and Raman spectroscopies, scanning electron (SEM), and optical microscopies to investigate their chemical composition and morphology, respectively. Water contact angle (WCA) analysis, dispersion and adhesion tests on porcine skin were also performed to evaluate the hydrophobic properties of all samples and to select the most suitable scaffold material for Naproxen drug delivery. Thermogravimetric analysis (TGA) was carried out to estimate the drug amount loaded into the diatomite, whereas the concentration of drug in vitro released to an artificial sweat buffer (EN 1811:2011) [40] was evaluated by fluorescence spectroscopy, recording the emission peak intensity of Naproxen at 360 nm. 

A further functionalization of the DF20 sample with PDA before (namely DF20_PDA) and after the drug loading (DF20_Nap_PDA) was carried out to evaluate if polydopamine may favour the adhesive properties of diatomite onto skin or it may improve the kinetics of drug release with respect to the uncoated DF20 analogue. Polydopamine is a melanin-like biopolymer easily obtainable via oxidative polymerization of dopamine under weak alkaline conditions. Its biocompatibility and adhesive properties make it suitable for embedding biohybrid systems or living microorganisms to let them adhere to specific substrates [22,41,42,43,44,45].

Transepithelial drug delivery was investigated by the deposition of drug-loaded diatomite samples on a 3D-organotypic tissue model EpiDerm^TM^ [46]. In particular, the successful trans-epidermal permeation of Naproxen over time was confirmed by observing, via high-performance liquid chromatography (HPLC), the released drug in the physiological medium in contact with the site of cells opposite to the diatomite deposition. After drug permeation, epithelial cell viability was demonstrated by MTT test and by observation of cells’ morphology, cytoskeleton and F-actin stress fiber organization by confocal microscopy.

### 2.1. Fourier Transform Infrared and Raman Spectroscopy of Diatomite Functionalized Samples

FT-IR spectra of all functionalized diatomite samples (Figure 1) show intense symmetric and asymmetric Si-O-Si stretching signals at 800 and 1100 cm^−1^, respectively, that are typical of silica-based materials [47]. The broad bands at 3450 cm^−1^ correspond to the O-H stretching peak of hydroxyl groups on the biosilica surface. In the presence of PDA, the signal at 3450 cm^−1^ is also more evident, being related to the presence of both the hydroxyl groups on the silica surface and the catechol moieties of polydopamine [41]. Moreover, the evidence of functionalization of diatomite surfaces with *n*-octyl groups in functionalized diatomite samples arises from the observation of C_sp^3^_-H stretching signals at 2923 cm^−1^ and 2863 cm^−1^. The signal intensity slightly increases by increasing the percentage of aliphatic chains from DF1 to DF40, and it is even higher for the DF20_PDA sample, which is likely due to the existence of the C_sp^2^_-H stretching signal in the same spectral region at ~3000 cm^−1^.

Results obtained via FT-IR were also strengthened by Raman analysis. Figure 2 shows the Raman spectra of DA, DF20 e DF40 in three different ranges reported in the same vertical scale: (a) 80–2000 cm^−1^, (b) 2600–3300 cm^−1^ and (c) 3300–4000 cm^−1^.

In the 80–800 cm^−1^ range, bands at ~300 cm^−1^ and ~370 cm^−1^ are due to the O-Si-O deformation, whereas the O-Si-O stretching is at ~400 cm^−1^. The signals at ~480 cm^−1^ and at ~600 cm^−1^ are due to the O_3_SiOH tetragonal vibration and to the (SiO)_3_-ring breathing, respectively, while the Si-O-Si stretching falls at ~700 cm^−1^ [48].

C-C stretching and C-H bending are evident at ~1468 cm^−1^, ~1530 cm^−1^ and ~1565 cm^−1^, while C-H and O-H stretching generate bands in the 2800–3100 cm^−1^ and 3400–3800 cm^−1^ ranges, respectively [49].

Spectra show similar signals before and after functionalization independently on the functionalization degree, but they significantly differ in a very narrow range, 1000–1450 cm^−1^, where bands appear due to C-C and C-H bonds. For DA, as shown in the black spectrum of the Figure 2a inset, weak bands at ~1034 cm^−1^, ~1096 cm^−1^, ~1350 cm^−1^, ~1384 cm^−1^ and ~1425 cm^−1^ can be attributed to C-H wagging, C-C stretching, C-H twisting and C-H bending of organic protoplasm residues [50]. On the contrary, in DF20 (blue line in the Figure 2a inset), some of these peaks tend to disappear and a broad band at ~1190 cm^−1^ is visible that becomes very intense and narrow in DF40 (red line) where it predominates at ~1210 cm^−1^ due to C-C stretching and/or C-H twisting. Spectra of DF1, DF5 and DF10 are not present in Figure 2 because they are similar to the DA spectrum, since Raman shows sensitivity to the percentage of functionalization by aliphatic chains starting from DF20.

Hence, the profiles of the Raman spectra in the 1000–1450 range confirm the successful surface functionalization of DF20 and DF40 samples by aliphatic chains.

### 2.2. Morphological Characterization

All diatomite samples were observed by both contrast phase (see Supporting Materials, Appendix A) and SEM microscopies (Figure 3) to evaluate and compare their morphology. Since fossil diatomaceous earth is highly dispersed, including biosilica shells and fragments from different species, observation was focused on valves with similar diameter lengths of ~50 µm. As expected, all nanostructures were not influenced by the presence of *n*-octyl chains on the surface: indeed, both meso- and nanopores were evident and unclogged either for the SEM micrographs of the bare activated diatomite or for those of the samples functionalized with alkyl chains and polydopamine (Figure 3 and Appendix A). Aggregation of biosilica debris was however observed for the DF20 and DF40 (Figure 3e,f) samples with the highest degree of aliphatic chains, as a possible effect of lipophilicity that induces attractive interactions of such biosilica fragments dispersed in the aqueous medium.

### 2.3. Static Water Contact Angle (WCA)

WCA analysis was carried out on dried spots of each DF sample and the results were compared with DA activated diatomite as the control. Analysis was also conducted for DF20_PDA to evaluate if the PDA coating could influence the hydrophobicity of the biosilica surface after functionalization with the *n*-octyl chains. 

As shown in Figure 4, hydrophobicity increases from the control DA (9°± 2) to the alkyl functionalized diatomite samples by increasing the percentage of functionalization, with DF20 and DF40 both showing the highest WCA value (~148°). These results agree with the dispersion tests in various solvents (see Supporting Materials, Appendix A), with DF1, DF5 and DF10 being more dispersible in polar solvents, whereas DF20 and DF40 are only dispersible in less polar solvents. Despite the well-known hydrophilic features of polydopamine, the diatomite sample coated with this biopolymer (DF20_PDA) showed a WCA value slightly higher than those of DF20 and DF40 (159° ± 5 versus 148° ± 7 and 148° ± 3, respectively). However, considering the standard deviations, the WCA value for DF20_PDA does not significantly differ from the DF20 one, meaning that PDA does not remarkably influence the overall hydrophobic behaviour of alkyl functionalized diatomite in water. 

### 2.4. Adhesion Test on Pig Skin

Adhesion tests on pig skin were performed to investigate the effect of functionalization on the adhesion properties of chemically modified DF samples. The adhesive properties of DF20_PDA and DA coated with PDA (namely DA_PDA) samples were also explored. To evidence the presence of biosilica samples deposited onto skin, before their deposition, we stained the samples with the orange-yellow emitting dye 4,7-di(thiophen-2-yl)benzo[c][1,2,5]thiadiazole [51] in acetone. After removal of the solvent by centrifugation, the stained biosilica samples were resuspended in hydroalcoholic EtOH/H_2_O 1:1 vol solution, spotted on porcine skin, and left to dry. When dried, the skin substrates were washed every 20 min for two hours and observed under UV light irradiation to evaluate the orange-yellow photoluminescence arising from the dye-stained diatomite still present on the skin surface after each washing step, as evidence of its adhesion. 

Figure 5 shows pictures of all diatomite samples deposited on the skin, including DA and the activated diatomite coated with polydopamine (DA_PDA) as the control. Results show that both control DA and DA_PDA samples are completely washed away after the second washing step, whereas DF20 and DF40 remained adherent for the whole experiment. On this ground, considering the lower degree of *n*-octyl groups for a similar adhesion output, DF20 was preferred to DF40 as the best candidate for drug delivery experiments. 

### 2.5. Drug Loading and Thermogravimetric Characterization of the Resulting Diatomite

Naproxen loading was carried out on both DF20 and control DA samples, by keeping diatomite in a hydroalcoholic solution of Naproxen (see Experimental) for 24 h at room temperature. The resulting drug loaded diatomite samples (namely, DA_Nap and DF20_Nap, respectively) were recovered after centrifugation, washed with bi-distilled water, and dried at 40 °C.

To estimate the drug amount loaded into the activated and the *n*-octyl (20%) functionalized diatomite, the thermogravimetric analysis was performed for both DA_Nap, DF20_Nap and control DA, DF20 samples. Thermograms shown in Figure 6 demonstrate a negligible weight loss for DF20 and DA because diatom biosilica has high thermal stability and the weight loss for DF20 related to the presence of *n*-octyl groups is expected to be very limited due to the low percentage of these alkyl chains versus the whole biosilica weight.

Conversely, a fast weight loss occurred at 250 °C for both DA_Nap (9.3%, i.e., 93 μg in 1 mg of diatomite) and DF20_Nap (11.2%, i.e., 112 μg in 1 mg of diatomite) samples as a result of the sole possible loss of Naproxen. Hence, 9.3% and 11.2% were, respectively, considered as the drug loading content in the DA_Nap and DF20_Nap samples investigated for drug delivery.

The higher amount of Naproxen in DF20_Nap versus DA_Nap also demonstrates that the *n*-octyl chains are not only suitable to favour the diatomite adhesion onto skin substrates, but they also promote the drug loading into the biosilica by their apolar affinity with the drug molecules that contributes to the drug retention in the porous scaffold.

### 2.6. In Vitro Naproxen Release Tests

First, drug delivery was investigated by recording over time the emission peak intensity at 360 nm of Naproxen released in an artificial sweat buffer solution where the drug-loaded samples were soaked under gentle stirring. Figure 7 shows the profile of drug released (% of release with respect to the drug-loaded amount) over time by DA_Nap, DF20_Nap, and both samples also coated with PDA (DA_Nap_PDA and DF20_Nap_PDA). The drug amount was interpolated from the calibration curve reported in the Supporting Materials (Appendix A). 

The best profile was found for DF20_Nap, gradually delivering the drug within 24 h, after which a plateau was observed (Figure 7, light orange circles) with a Nap release of up to 80%. Conversely, DA_Nap fully released the drug immediately after its immersion in the sweat buffer (Figure 7, light green circles). This result leads to the consideration that the *n*-octyl chains on the DF20 surface were suitable to interact with the loaded drug molecules, slowing down their diffusion into the aqueous medium [52,53].

Moreover, the PDA coating was found to inhibit drug delivery. Indeed, both DF20_Nap_PDA and DA_Nap_PDA released less than 5% of the drug with respect to the DF20_Nap analogue after 72 h. Hence, the presence of the PDA coating layer was not considered adequate for our aim.

To elucidate the release mechanism of Naproxen, the Korsmeyer–Peppas model was applied as follows:$$\frac{M_{t}}{M_{\infty }}=Kt^{n}$$
with $M_{t}$ and $M_{\infty }$ indicating the cumulative release of drug at time *t* and infinity, respectively, and *K* is the proportionally constant while *n* is the release index. The data obtained are shown in Table 1 and the *n* value close to 0.5 shows that the drug release mechanism follows the Fick’s law for samples DF20_Nap [53]. 

### 2.7. Effect on Living Human Epidermal Tissue Viability of Drug-Loaded Diatomite 

All diatomite samples loaded with Nap (DA_Nap, DF20_Nap, DA_Nap_PDA and DF20_Nap_PDA) were kept in contact with EpiDerm™ inserts for 24 h only at the apical side, thus mimicking a topical treatment in vivo. Samples (1 mg) were resuspended in 70 µL of cell culture medium and placed on the apical side of the human tissues, while the bottom chamber of each well was filled with 1 mL of culture medium. 

Tissues were also exposed to free Naproxen, whereas a positive control of viability was represented by the sole tissues exposed to the culture medium at the apical side. Tissue viability was evaluated with MTT assay. As reported in Figure 8, no statistically significant changes in cell viability were observed in any of the experimental conditions. This result suggests that Naproxen and diatomite do not cause cytotoxicity when applied topically.

### 2.8. Effect on EpiDerm^TM^ Actin Cytoskeleton upon Exposure to Drug-Loaded Diatomite 

The possible effects of drug-loaded diatomite samples on cell cytoskeleton organization were also investigated. After exposure experiments, tissues were fixed and subjected to analysis to visualize the actin-based cytoskeleton using fluorescent phalloidin.

Confocal microscopy of EpiDerm™ exposed to the sole medium revealed a cellular compact morphology, well organized F-actin stress fibers within the cytoplasm of keratinocyte (Figure 9). None of the treatments with the different Naproxen-loaded diatomite samples induced appreciable changes in the organization of the cytoskeleton versus the control cells. The observation of both cell morphology and organization of the actin cytoskeleton, in agreement with the cell viability data, indicated the absence of alterations related to the cytotoxic effect.

### 2.9. Trans-Epidermal Drug Permeation Tests

As for the viability assessment, each drug-loaded diatomite sample (1 mg) was resuspended in 70 µL of cell culture medium and deposited on the apical side of human tissues. Considering the Naproxen amount in diatomite evaluated by TGA analysis, a reference Nap sample of the free molecular drug in solution was prepared dissolving 100 µg of Naproxen in the culture medium. This solution was kept in contact with the apical side of the tissue as a control trans-epidermal test. A further negative control (CTRL) was prepared, with the culture medium in contact with both sides of the tissue sample. At different time intervals of exposure (1, 3, 5, 9 and 24 h), 50 µL of basolateral culture media were collected to evaluate the concentration of permeated drug via HPLC. Each withdrawn sample was replaced with an equivalent amount of fresh culture medium to keep the overall volume constant.

As shown in Figure 10, permeation of free Nap through tissues is faster than that from DA_Nap and DF20_Nap. Moreover, the observation of a gradual trans-epidermal permeation (light green squares in Figure 9) of the drug from DA_Nap with respect to the very fast delivery from the same sample in artificial sweat (green square profile in Figure 6), suggests that the bare diatomite DA is not really efficient as a drug delivery system because it quickly delivers the drug in the apical solution in contact with EpiDerm^TM^ and, thereafter, the free released drug permeates the human tissue. Conversely, the results recorded for DF20_Nap both in artificial sweat (light orange circles in Figure 6) and by deposition on EpiDerm^TM^ (light orange circles in Figure 9) are coherent, evidencing in both cases a good control of delivery and highlighting the suitability of the DF20 diatomite in releasing Naproxen gradually over time.

According to the in vitro experiments in artificial sweat buffer, trans-epidermal permeation of the drug delivered by polydopamine-coated samples (DA_Nap_PDA and DF20_Nap_PDA) was very limited: the drug concentration in the basolateral chamber was lower than the HPLC limit of detection even 24 h after the beginning of permeation tests.

## 3. Discussion

Mesoporous silica is an extremely profitable material for drug delivery and biomedicine applications [54]. Indeed, therapeutic approaches availing of silica-based systems generally provide positive biological responses by bone and cartilage tissues due to the biocompatibility and surface morphology of mesoporous silica [55]. In particular, silica nanoparticles (SNPs) deserve interest for drug delivery in skin tissues [56], although they suffer from a massive uptake by cells, with related intrinsic toxicity and not well-clarified scavenging processes of the SNP residues. The use of mesoporous micrometre-size silica particles can overcome any side effect related to uptake by cells. 

For our study, we have selected fossil biosilica shells from diatom microalgae (diatomaceous earth, DE) as a green, abundant, low-cost source of polydisperse silica microparticles with high surface area, mesoporosity and biocompatibility with a great variety of isolated cell lines [57]. DE also shows biological properties, such as the long-lasting release of Si-rich compounds (e.g., silicic acid) that stimulate the activation of living mammalian cells important for wound healing processes [58].

Here we report for the first time, a diatom biosilica-based trans-epidermal delivery system of Naproxen, selected as a model NSAID drug for skin tissues. DE was previously purified via an acid-oxidative protocol. The biosilica surface was then functionalized with *n*-octyl groups to favour their hydrophobic interaction with the lipid matrix of the skin, allowing adhesion. The resulting samples were characterized by FT-IR and Raman spectroscopy (Figure 1 and Figure 2) and SEM microscopy (Figure 3). WCA analysis (Figure 4) and adhesion tests to porcine skin (Figure 5) show that, among the various functionalized diatomite samples that differ in the alkyl chains surface density, DF20 functionalized with 20% (*w*/*w*) triethoxy-octylsilane versus diatomite is the best candidate for drug delivery.

Hence Naproxen was loaded in DF20, and the loading amount was estimated by thermogravimetric analysis (Figure 5) as 11.2% of the sample weight.

Drug delivery was investigated in vitro by comparing the efficiency of DF20_Nap sample with the drug-loaded, activated diatomite DA_Nap used as the control, to evaluate the effect of surface aliphatic chains on the delivery efficiency of DF20. Samples were dispersed under continuous stirring in an artificial sweat buffer, and the concentration of Naproxen dissolved in solution was measured by spectrofluorimetric analysis at different time intervals. Results show that the chemical surface modification with octyl groups allows not only the DF20 adhesion to skin substrates but also the gradual release in 24 h of the drug into an artificial aqueous medium mimicking the sweat of human skin. The *n*-octyl groups are thought to retain the drug by hydrophobic interactions inhibiting its diffusion in water. Conversely, drug delivery from the bare diatomite DA was very fast, occurring immediately after the beginning of the experiment.

The 24 h time delivery observed for DF20 before reaching a Nap concentration plateau, is in agreement with the result reported in the literature for ad hoc synthesized silica nanoparticles [52,53], functionalized with aminopropyl triethoxysilane with the aim of controlling the drug delivery rate by electrostatic interactions of amino groups on nanoparticles with carboxyl groups of Naproxen. However, this SNP-based drug delivery system was only studied for oral administration application and no evidence of adhesion properties on the skin as well as no investigation of effects on biological tissue viability were reported. The same remark can be extended to the work reported by Aw et al. [21], where diatom frustules were suggested for Indomethacin delivery by oral administration, with no investigation about their adhesion properties for topical use or biocompatibility versus the skin.

We also explored the possible effects of a polydopamine [42,59,60] coating layer on the DF20_Nap surface, but no remarkable difference in terms of WCA and adhesion on porcine skin was observed with respect to the uncoated analogue sample. Moreover, polydopamine significantly inhibited the Naproxen delivery, in terms of both the overall amount and rate of the drug released in vitro.

Besides the common drug delivery tests in aqueous media, we also investigated, for the first time, the trans-epidermal permeation of Naproxen from DF20_Nap through an organotypic model of the human epidermis (EpiDerm^TM^), to have a more accurate representation of the in vivo cellular environment, including cell–cell and cell–matrix interactions, which can influence the behaviour and function of the human epidermis.

The EpiDerm^TM^ model tissue can well represent the barrier function of human skin, which has a complex structure made of multiple layers with distinct cell types. Organotypic models provide a more reliable platform for testing the effects of drugs, drug permeability and water loss on the human epidermis. By considering the three-dimensional architecture and cellular composition of the tissue, these models can better predict drug permeation, efficacy and toxicity compared to traditional bidimensional 2D cell cultures. This can aid in the development of safer and more effective treatments targeting human epidermis-related conditions.

Results of the trans-epidermal permeation of Naproxen from DF20_Nap through EpiDerm^TM^ were in accordance with those observed for drug delivery in vitro from the same diatomite sample to the sweat buffer. Conversely, the same experiments carried out on the DA_Nap sample evidenced that the bare diatomite quickly delivers the drug in aqueous solution, as shown in Figure 6, and thereafter the free drug molecules gradually reached the basolateral chamber of the skin tissue cells (Figure 9).

We also carried out trans-epidermal permeation tests for the polydopamine-coated sample DF20_Nap_PDA as well as for the in vitro drug delivery test, but the drug release over time was not successful, since the Naproxen concentration in the basolateral chamber was not detectable by HPLC. This may have important implications for those formulations whose aim is to limit the location of drugs in the epidermis, reducing transdermal permeation and thus achieving a specific local effect on the epidermis.

No statistically relevant variation in tissue viability was observed by MTT assay after incubation of EpiDerm^TM^ with all the drug-loaded diatomite samples, as evidence of the biocompatibility of diatomite and Naproxen with skin cell tissues.

Moreover, confocal microscopy of EpiDerm™ exposed to drugs and different treatments revealed a cellular compact morphology and well-organized F-actin stress fibers within the cytoplasm of keratinocytes with no differences with respect to the control.

In summary, our study provides a proof of concept that proper chemical functionalization of diatomite with hydrophobic aliphatic chains is a profitable strategy to develop biohybrid drug delivery systems showing both good adhesion properties to the lipidic matrix of the skin surface and efficient control of drug release over time. These systems could represent an eco-friendly and cheap alternative, or they could be envisaged as implementing materials for spray, hydrogels or patches already tested for skin therapies, with possible long lasting drug delivery efficiency.

The investigation carried out for topical administration of Naproxen selected as a model anti-inflammatory drug is in principle extendable to other kinds of pharmacologically active molecules, thus paving the way to a wide series of diatom biosilica-based systems for topical drug delivery. The evidence of biocompatibility and trans-epidermal drug permeability through an organotypic model of the human epidermis, investigated herein for the first time, even strengthens the concept that tailored chemically functionalized diatomite-based materials, such as the DF20 sample, are eligible candidates for drug delivery in skin treatment diseases.

## 4. Materials and Methods

### 4.1. Materials

Diatomaceous earth, H_2_SO_4_ 98%, H_2_O_2_ 30%, triethoxy-octylsilane, potassium bromide, Naproxen sodium and all solvents (including ethanol, water and acetonitrile used at HPLC grade) were purchased from Sigma Aldrich (Darmstadt, Germany). The reconstructed human epidermis models (Epiderm™, EPI-200) were obtained from Mattek Corp. (Ashland, MA, USA). Alexafluor 488 Phalloidin (cat. #A12379, dilution 1:500), paraformaldehyde, Triton X-100, bovine serum albumin, *n*-propylgallate and propidium iodide were purchased from Thermo Fischer Scientific, Waltham, MA, USA. 

### 4.2. DE Purification and Chemical Modification

According to our previously reported protocol [57], DE (7.0 g) was suspended in bi-distilled water (30 mL) and stirred at 70 °C inside a 100 mL round bottle flask. Then, H_2_SO_4_ (98%, 5 mL) was added, and the suspension was stirred for 10 min. A further dropwise addition of H_2_O_2_ (30%, 5 mL) was subsequently performed and the reaction mixture was kept under stirring for 4 h at 70 °C. After cooling at room temperature, a series of washing steps with bi-distilled water were performed to remove acids. Finally, the activated biosilica sample DA was isolated collecting the pellet after centrifugation at 3000 rpm for 10 min and by drying it in a stove at 40 °C.

### 4.3. Chemical Functionalization of the Activated DA Surface

Experiments were carried out to functionalize DA with *n*-octyl chains at different surface densities, with the aim of tuning the hydrophilic nature of the diatomite surface. In particular, DA (1 g) reacted with triethoxy-octylsilane (TeC_8_S:DA 1, 5, 10, 20 and 40% *w*/*w*, respectively) according to a slightly modified chemical functionalization method from the literature [47]. In a 100 mL round bottle flask equipped with a reflux condenser, DA (1 g) was suspended in toluene (5 mL), stirred for 20 min at 90 °C and then bi-distilled water (40 μL) was added. The suspension was stirred for a further 20 min. Then, TeC_8_S was added, and the mixture was kept under continuous stirring. After 4 h, it was cooled and washed in a beaker three times with acetone and ethanol. The resulting product, (namely DF1, DF5, DF10, DF20 and, DF40 in dependence of the functionalization degree) was eventually collected and dried at 40 °C. 

### 4.4. Characterization of Diatomite Samples

#### 4.4.1. Fourier Transform Infrared Spectroscopy (FT-IR)

FT-IR spectra of DF samples were recorded in the presence of dry KBr by an INVENIO-S Bruker FT-IR spectrometer (Bruker Optics, Ettlingen, Germany) with 4 cm^−1^ resolution in 4000–400 cm^−1^ wavenumber range for 36 scans.

#### 4.4.2. Raman Spectroscopy

Raman spectra were recorded by a LabRAM HR (Horiba-Jobin Yvon, Montpellier, France) spectrometer with 532 nm excitation laser under ambient conditions. Low laser power (<1 mW) was used to avoid heat-induced modification or degradation of the sample due to focused laser light during spectrum acquisition. The collection time was longer than 200 s. The excitation laser beam was focused through a 50× optical microscope (spot size 1.3 mm, working distance 1 cm). The spectral resolution was 1 cm^−1^.

Raman measurements were used to detect vibrational bonds of Si with O, C with H and C and O with H both to obtain a fingerprint of diatoms and to test their functionalization by aliphatic chains.

#### 4.4.3. Static Water Contact Angle (WCA)

WCA measurements were carried out with 2 μL water droplets using a KSV CAM 200 instrument. Tests were made after drop-casting a series of 1 mg/mL sample dispersions in a water and ethanol (1:1) solution until a homogenous sample spot on a glass slide was obtained. Each WCA value represents the average value of three tests. 

#### 4.4.4. Adhesion Test

The adhesion ability on porcine skin substrates was evaluated for DF1, DF5, DF10, DF20, DF40 and DF20_PDA versus the control samples DA and DA_PDA. In all cases, preliminary staining of biosilica was performed with an orange emitting dye of our synthesis [4,7-di(thiophen-2-yl)benzo[c][1,2,5]thiadiazole]. A total of 100 μL of a solution of the dye in acetone (1 mg/mL) was added to 1 mL of a mixture of water and ethanol (1:1 vol) [50]. Then, diatomite (20 mg) was added and stirred overnight at room temperature to allow the staining. The resulting stained samples were washed three times with fresh hydroalcoholic mixture (3 × 1 mL) and eventually resuspended in 1 mL of the same fresh solution. Porcine skin was purchased from a local seller and immediately used. The epidermis was dissected from the fatty tissue, washed, and cut into 100 cm^2^ squares. Spots of 100 µL were dropped on a 4 cm^2^ skin surface and left to dry. Once dried, the skin substrates coated with DF samples were subjected to flushing with bi-distilled water every 20 min for two hours and, after each washing step, the photoluminescence of the dye-stained diatomite was checked under UV irradiation, as qualitative proof of the adhesion resistance of samples onto the skin surface.

#### 4.4.5. Thermogravimetric Analysis (TGA)

TGA analysis was carried out for DA, DA_Nap, DF20 and DF20_Nap by a Perkin Elmer Thermogravimetric Analyzer Pyris 1 (Shelton, CT, USA), using platinum pans, 10 °C/min heating rate and 30 mL/min N_2_ flux. 

### 4.5. Naproxen Loading into Diatomite Samples

The diatomite sample DF20 (20 mg), selected as the best candidate for the drug delivery test, was added to a solution of Nap (70 mg, 50 mM) in a mixture of ethanol and water (6 mL, 60% *v/v*) and kept under continuous stirring for 24 h. The same procedure was carried out for DA (20 mg) as the control. The drug-loaded diatomite samples, called DA_Nap and DF20_Nap were collected by centrifugation at 3226× *g* washed three times with bi-distilled water and dried at 40 °C for 24 h.

A further drug-loading experiment was repeated for DF20 (20 mg), with the aim of coating the resulting sample with a layer of polydopamine (PDA). Namely, a DF20_Nap_PDA sample was obtained upon oxidative polymerization, on the surface of the drug-loaded diatomite [41], of dopamine (2 mg/mL) in TRIS buffer (1 mL; pH 8.2, 10 mM) under gentle stirring for 3 h. The resulting DF20_Nap_PDA product was collected and dried at 40 °C after centrifugation and several washing steps with TRIS buffer.

### 4.6. In Vitro Naproxen Delivery Test

Drug delivery was investigated by stirring the drug-loaded samples (5 mg) soaked in the artificial sweat buffer (10 mL) reported for the EN 1811:2011 reference test [40]. During incubation, 2 mL of buffer was collected every hour, centrifuged, and analysed by an ALT Varian Cary Eclipse fluorimeter (Agilent, Santa Clara, CA, USA), to record the emission peak of Naproxen at 360 nm, as proof of its delivery in solution. The collected fractions were rejoined in the corresponding batches after each analysis to ensure sink conditions. 

### 4.7. Naproxen Trans-Epidermal Permeation Test and Tissue Viability

Nap trans-epidermal permeation test and tissue viability were carried out with EpiDerm^TM^ (EPI-200), a human 3D tissue model obtained from Mattek Corp (Ashland, Massachusetts). This artificial tissue consists of human-derived epidermal keratinocytes (NHEK), cultured at the air–liquid interface on a semi-permeable tissue culture insert to recreate a multilayered, highly differentiated model of the human epidermis. The tissue model exhibits organized basal, spinous, granular layers, and cornified layers mimicking the human epidermis in vivo. The drug-loaded samples (DA_Nap, DF_Nap, DA_Nap_PDA and DF_Nap_PDA) were resuspended in 70 µL of cell culture medium and placed on the apical side of human tissues, while the bottom chamber of each well was filled with 1 mL of culture medium. Grounding on TGA results, the control Nap solution was prepared dissolving Naproxen (100 µg) in the culture medium (70 µL), to have a similar concentration of the drug with respect to the other samples. A tissue negative control (CTRL) was prepared, with sole culture medium in both sides of the sample. Experiments were performed at 37 °C keeping the samples in an incubator. At different time points (1, 3, 5, 9 and 24 h) after the exposure, 50 µL were withdrawn from the basolateral compartment and replaced with an equivalent volume of fresh culture medium. Every aliquot was immediately stored at −20 °C until HPLC analysis. Considering the dilution due to the sampling procedure, drug concentration values (C_c_) were corrected with the following equation [61]:$$C_{C}=V_{r}C_{t}+V_{s}\sum_{i=1}^{1-n}C_{i}$$

*C_t_* = drug concentration at sampling time;

*C_i_* = drug concentration for *i^th^* time;

*V_r_* = volume of the receiving chamber;

*V_s_* = removed volume for analysis.

### 4.8. Tissue Viability MTT Assay

The effect of drug and diatomite samples on EpiDerm™ viability was assessed by the MTT tissue viability assay.

After transport at 4 °C, the packaging agarose was removed and tissues were re-equilibrated overnight into a humidified 37 °C, 5% CO_2_ incubator by placing inserts in 6-well plates containing 1 mL per well of assay medium (provided by MatTek and included with the tissues).

Following the re-equilibration period, tissues were apically exposed to 70 µL of drug-loaded diatomite samples while the bottom chamber of each well still contained 1 mL of culture medium. Control tissues maintained the culture medium in the apical compartment. After 24 h exposure to drugs, the EpiDerm™ inserts were washed with phosphate-buffered saline (PBS) solution to remove any remaining applied material.

Then, tissues were placed in a 24-well plate containing 300 μL of medium with 1 mg/mL solution of MTT (3-(4, 5-dimethylthiazolyl-2)-2, 5 diphenyltetrazolium bromide) and they were incubated for 3 h at 37 °C, 5% CO_2_. 

During the reaction, the yellow tetrazolium salt MTT is converted to purple formazan crystals by intracellular reducing equivalents produced by metabolically active cells. Subsequently, tissues were submerged in 2 mL of isopropyl alcohol for 2 h under shaking to dissolve the generated formazan crystals. The spectrometric absorbance in each well was read at 595 nm using a microplate reader (Bio-Rad, Hercules, CA, USA). The viability of tissues treated with the different formulations was expressed as a percentage of the mean of control tissues.

### 4.9. Immunofluorescence and Confocal Microscopy

After each treatment, tissues were fixed with 4% paraformaldehyde in phosphate-buffered saline (PBS) for 1 h at room temperature and washed twice for 5 min in PBS. Tissues were permeabilized with 0.1% Triton X-100 in PBS for 15 min at room temperature followed by an additional three washes in PBS.

Tissues were blocked with 1% bovine serum albumin in PBS for 45 min at room temperature and incubated with Alexafluor 488 Phalloidin for 1 h at room temperature. All incubations were performed from both sides of the filters.

After three washes in PBS, filters were excised from the support, mounted in PBS/glycerol (1:1) containing 1% *n*-propylgallate, pH 8.0 on microscope slides and confocal pictures were taken with a Leica SP5 microscope (Leica SP5, Leica Microsystems, Milan, Italy).

### 4.10. Determination of Nap Trans-Epidermal Permeation by HPLC

HPLC analysis was performed with a reverse-phase Supelco LC-18 column (15 cm length, ID 6 mm, 5 μm particle size) with a mixture of methanol, water, and acetonitrile in 80:17.5:2.5 volumetric ratio as the mobile phase. The pH of the mobile phase was kept at 3 by adding orthophosphoric acid. Analyses were performed at 40 °C oven temperature, in isocratic mode with 1 mL/min flow rate. The chromatographic signals were recorded by a PDA detector at 272 nm wavelength. 

## Figures and Tables

**Figure 1 marinedrugs-21-00438-f001:**
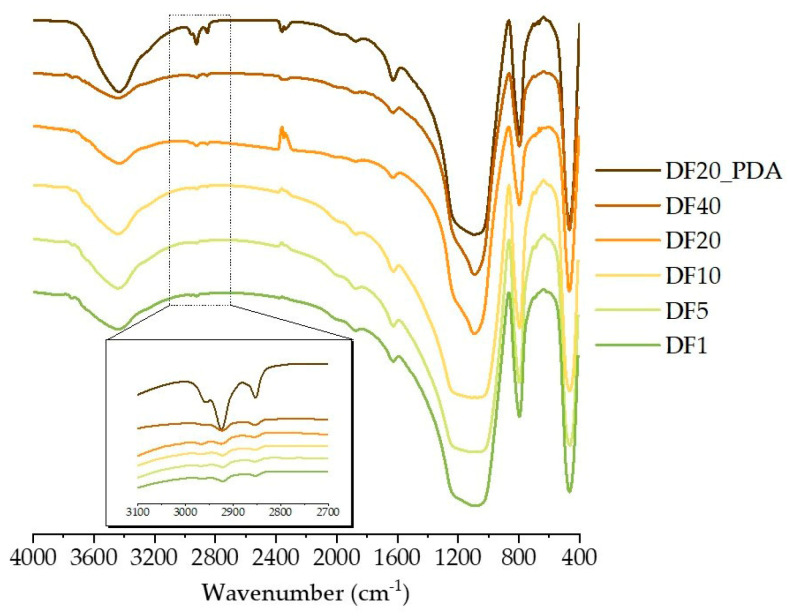
FT-IR spectra for DF1, DF5, DF10, DF20, DF40 and DF20_PDA. Inset: magnification of 2700–3200 cm^−1^ spectral region.

**Figure 2 marinedrugs-21-00438-f002:**
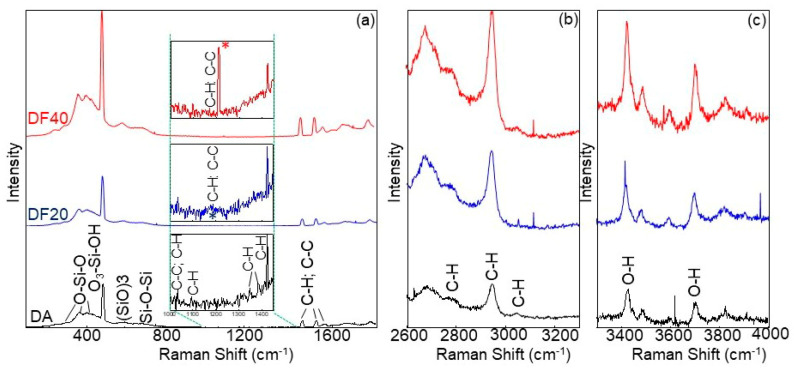
Raman spectra of DA (black line), DF20 (blue line) and DF40 (red line) in three different ranges: (**a**) 80–2000 cm^−1^, (**b**) 2600–3300 cm^−1^, and (**c**) 3300–4000 cm^−1^. Insets in (**a**) show the corresponding 1000–1450 cm^−1^ zoomed area, with blue and red asterisks indicating additional bands due to C-H and C-C bonds that appear after functionalization. Bonds involved in Raman bands are also reported.

**Figure 3 marinedrugs-21-00438-f003:**
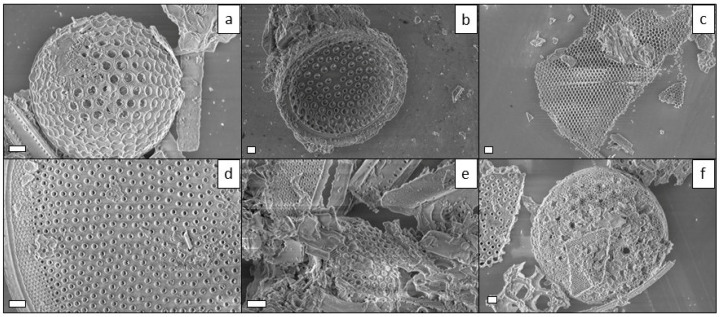
SEM micrographs of (**a**) DA, (**b**) DF1, (**c**) DF5, (**d**) DF10, (**e**) DF20 and (**f**) DF40. Scale bars: 2 µm.

**Figure 4 marinedrugs-21-00438-f004:**
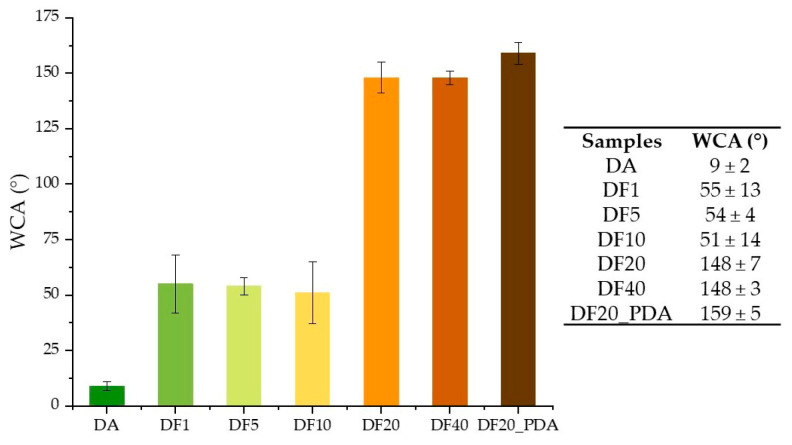
Histogram with WCA values (listed in the table) of DA, DF1, DF5, DF10, DF20, DF40 and DF20_PDA.

**Figure 5 marinedrugs-21-00438-f005:**
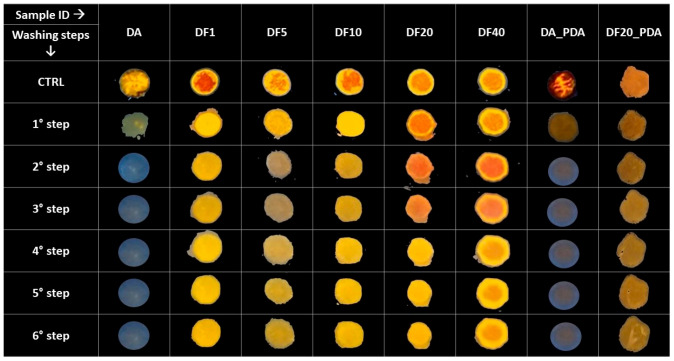
DA, DF1, DF5, DF10, DF20, DF40 and DF20_PDA spots on pig skin excited by a UV light at 365 nm after every washing step.

**Figure 6 marinedrugs-21-00438-f006:**
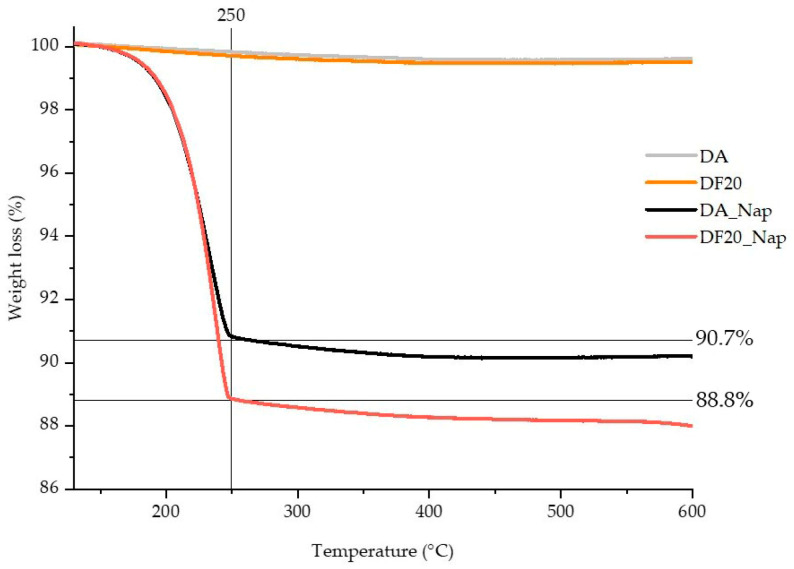
Thermograms for DA, DA_Nap, DF20 and DF20_Nap.

**Figure 7 marinedrugs-21-00438-f007:**
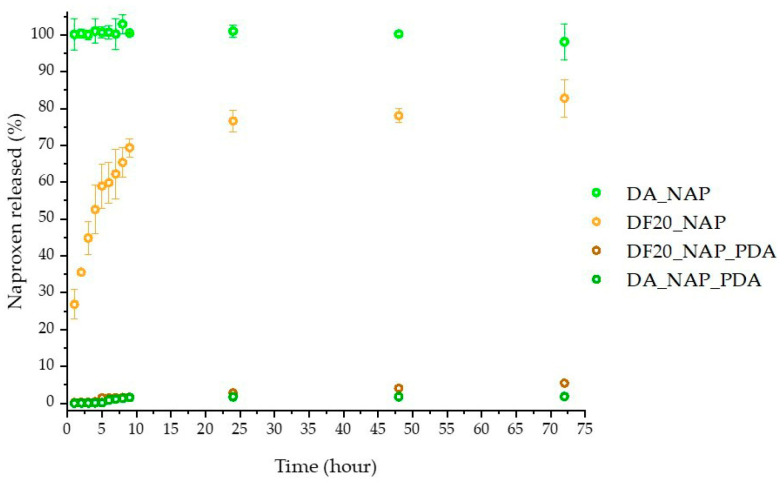
Naproxen time-released profile for DA_Nap, DA_Nap_PDA, DF20_Nap and DF20_Nap_PDA.

**Figure 8 marinedrugs-21-00438-f008:**
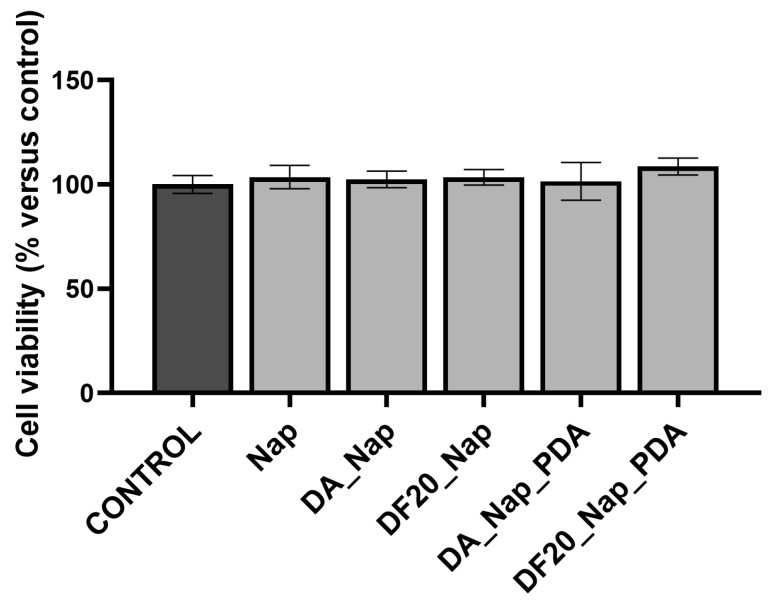
Viability of EpiDerm™ tissues apically exposed to medium and diatomite samples for 24 h. Tissue incubated with the medium was used as control, and their viability was set to 100%.

**Figure 9 marinedrugs-21-00438-f009:**
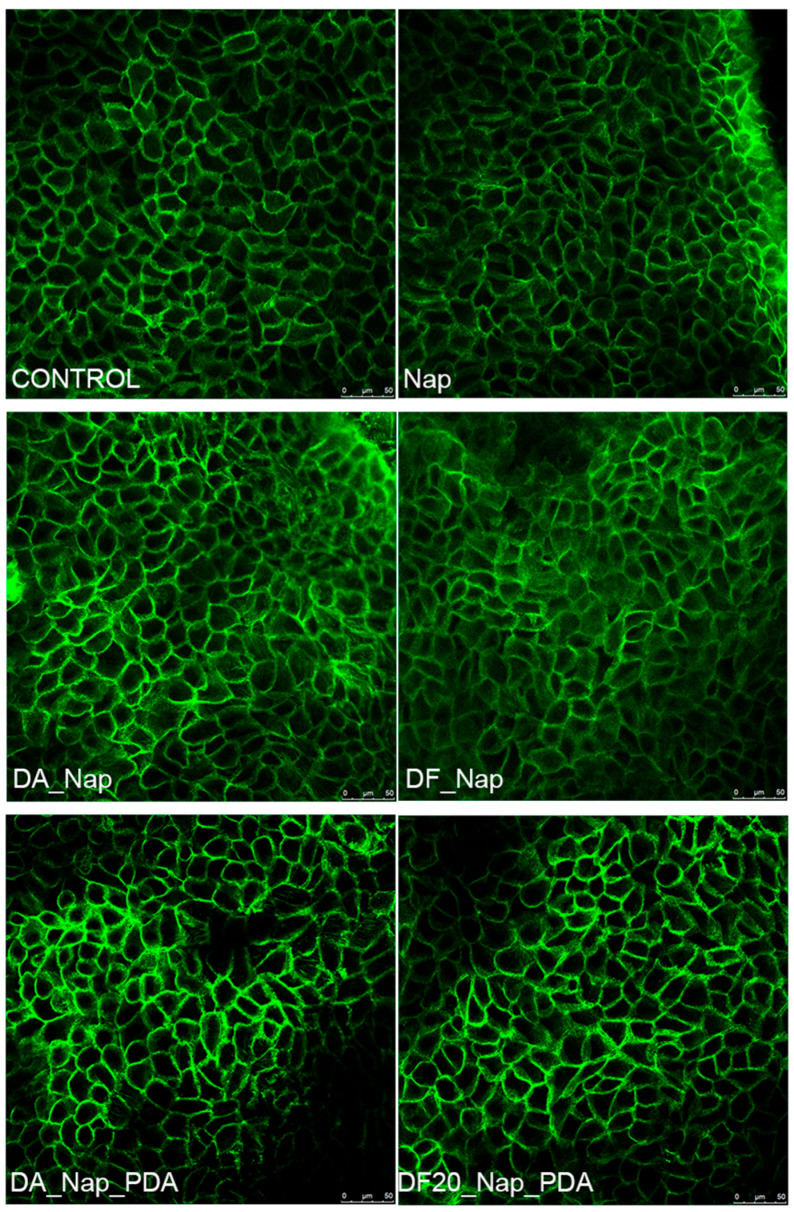
EpiDerm™ tissues apically exposed to medium and tested with DA_Nap, DF20_Nap, DA_Nap_PDA and DF20_Nap_PDA samples for 24 h, fixed in PFA and subjected to immunofluorescence. The actin cytoskeleton was visualized by incubating the tissues with fluorescent phalloidin. Scale bars = 50 µm.

**Figure 10 marinedrugs-21-00438-f010:**
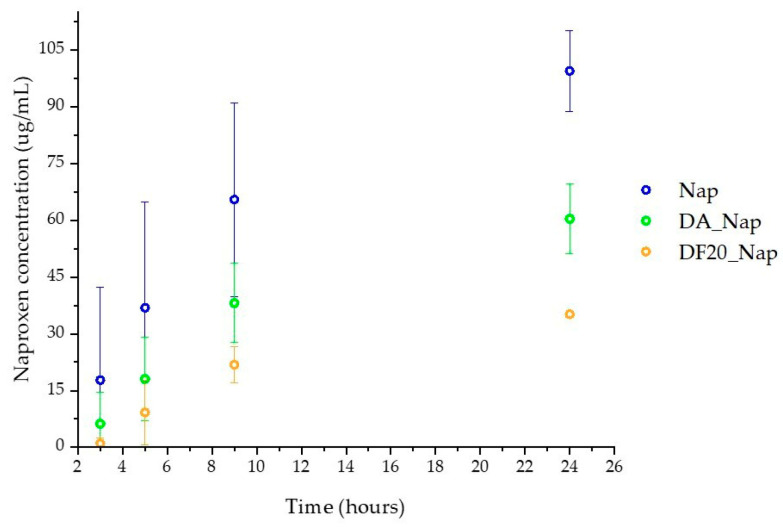
Trans-epidermal release profile across EpiDerm^TM^ of Nap, DA_Nap and DF20_Nap as µg/mL of drug released in the basolateral chamber. DA_Nap_PDA and DF_Nap_PDA are not shown since the permeated drug concentration was lower than the HPLC detection limit.

**Table 1 marinedrugs-21-00438-t001:** Kinetic parameters obtained by the Korsmeyer–Peppas mathematic model for DF20_Nap.

Sample	K	n	R^2^
DF20_Nap	26.69	0.47	0.993

## Data Availability

Additional data are available in the Supporting Information.

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
