# Peer review of "Drug Delivery through Epidermal Tissue Cells by Functionalized Biosilica from Diatom Microalgae"

_marinedrugs, 2023, doi:10.3390/md21080438_

Round 1
Reviewer 1 Report
In this work, authors explained the capacity of functionalized diatomaceous earth to be used as systems for the transdermal delivery of naproxen as a proof of concept. Their work is an interesting approach, however, for publication of their results, authors must deeply improve the manuscript. Some suggestions are explained bellow:
- Firstly, I recommend the author to check all the misprints along the text, as in line 97 (diameter), for example.
- The paragraph 101-105 seems to me more a conclusion or a hypothesis than an introduction. I suggest authors to change it to the end of the introduction section, for example.
- In line 117, the HLB requirements for the drugs to be able to penetrate the skin is not good explained, as that explanation could be for every route of administration. As authors should know, an adequate HLB is a key step for the absorption and distribution of every active. Please, improve it.
- Authors applied a TGA test for the estimation of the loading of naproxen. I do not really think this a correct test for that purpose, maybe as a support test, but no the principal and only one. As the DE were loaded in naproxen solution, I would suggest the measurement of that solution with an analytical test (as the flourescence test, HPLC...), and then, carried out the estimation with the difference value, from the initial solution to that one.
-Could the authors explain if they run a RAMAN test or any molecular characterization of the DE and their functionalization, besides FTIR?
- For the adhesion test, I will recommend authors to carry out a mechanical test with a texturometer, to compare quantitatively the force with and without the functionalization.
- In the release results (line 268-274) authors must deeply improve the explanation and graphs. Firstly, I will suggest the authors to show their results as % amount compared with the initial loading, as it is easier for the reader to understand. Secondly, the release process is not a race classification, so, say "the tenth" or "the hundredth" is completely incorrect. In those cases, you have to indicate the result as "the 10% or the 100%" of the drug was released", for example. Could you fit those deliveries to a kinetic model, please? Is there any statistical differences between the samples?
- Regarding the cell viability assay, please, indicate in the Y-axis that values are compared with the control (% versus control).
English have some misprints. In general, is correct.
Reviewer 2 Report
1. The author should check the terms ant their abbreviations carefully. They are extremely confusing. For example, in the 13th line diatomaceous earth (DE), diatomaceous earth (DA) in the 140th line. What is the DF, DA_PDA, DA_Nap, DF20_Nap, DF20_Nap_PDA? When they first appear, they should be given a clear indication.
2. A clear scale bar should be added to the images of Figure 2.
3. Since the PDA coating layer was not profitable for delivering Nap, what is the purpose of introducing PDA in this study?
4. Why did not use HPLC to measure the drug loading of Nap in samples?
5. If the Naproxen-loaded mesoporous biosilica is used for therapies of skin diseases, what are the advantages and disadvantages of DF20_Nap compared with other drug formulations including hydrogels, sprays? It should be discussed in the manuscript.
6. The author should build a dermal-disease model to verify the efficacy of the DF20_Nap.
No
Reviewer 3 Report
In this manuscript, the authors reported diatom-based functionalized biosilica for drug delivery through epidermal tissue cells. This work seems to be useful for this field. However, the following problems should be addressed before further consideration of publication:
1. The keywords should be reduced, and five typical ones are enough.
2. A scheme can be created after the Introduction section to better demonstrate the contents and innovations.
3. The Introduction is suggested to be concise to come straight to the point, and references should be enriched.
4. High-resolution figures are needed, and the insets should be clear. All the figures need to be revised with improved quality, where consistent layout/size are desired to improve the readability.
5. Figure 3 is confusing and the detailed explanation is needed. In general, PDA coating is used to enhance the hydrophilicity.
6. Figure 6 and Figure 9 are suggested to be optimized. Maybe some other styles are suitable for demonstration.
7. The development and novelty of this work can be stated, especially the superiority or enhancement when compared with other advances. What do you think can further improve the performance, and is it feasible for future practical applications?
Round 2
Reviewer 1 Report
Authors have change and clarify all the concerns I had in the first read. I want to congratulate authors for the improvements made in the manuscript. At this point, I think is ready to be published. Happy Holidays!
English is ok.
Reviewer 2 Report
No.
Reviewer 3 Report
I have checked all the revisions. The revised manuscript has improved most issues, yet the following problems still need to be addressed:
1. For better readability, well-designed figures are needed, which should be created in compact layout and consistent word size. You can recreate the figures refer to published papers in Marine Drugs (e.g., 10.3390/md21050266).
2. The Introduction is suggested to be revised with merged paragraphs, and the related references of drug delivery should be added including: 10.1021/acsami.1c16859, 10.1002/EXP.20220147.